# Postoperative Chylothorax after Modified Radical Neck Dissection for Thyroid Carcinoma: A Missable Rare Complication of Thyroid Surgery

**DOI:** 10.3390/medicina56090481

**Published:** 2020-09-21

**Authors:** Junghyun Lee, In Eui Bae, Jin Yoon, Keunchul Lee, Hyeong Won Yu, Su-jin Kim, Young Jun Chai, June Young Choi, Kyu Eun Lee

**Affiliations:** 1Department of Surgery, Seoul National University Bundang Hospital, Seongnam-si, Gyeonggi-do 13620, Korea; 54299@snubh.org (J.L.); genijin@gmail.com (J.Y.); 82295@snubh.org (K.L.); hyeongwonyu@gmail.com (H.W.Y.); 2Department of Surgery, Konyang University Hospital, Daejeon 35365, Korea; inniggo@gmail.com; 3Cancer research institute, Seoul National University College of Medicine, Seoul 10408, Korea; su.jin.kim.md@snu.ac.kr (S.-j.K.); kyueunlee@snu.ac.kr (K.E.L.); 4Department of Surgery, Seoul National University Hospital, Seoul 03080, Korea; 5Department of Surgery, Seoul National University Boramae Medical Center, Seoul 07061, Korea; kevinjoon@naver.com

**Keywords:** postoperative chylothorax, postoperative complication, thyroidectomy, modified radical neck dissection, papillary thyroid cancer

## Abstract

*Background and objectives*: Currently, few studies have been conducted on postoperative chylothorax, specifically in total thyroidectomy with modified radical neck dissection (MRND) in papillary thyroid carcinoma patients. This study provides the actual incidence, etiology, and clinical features of postoperative chylothorax and reports the clinical outcomes after treatment, which were dependent upon the severity of the complications. *Materials and Methods*: The medical charts of 111 papillary thyroid cancer (PTC) patients who underwent total thyroidectomy with modified radical neck dissection from January 2016 to December 2018 were reviewed retrospectively. The results were compared in three groups: the no chylothorax group, the subclinical (asymptomatic) group, and the clinical (symptomatic) group. *Results:* Chylothorax occurred in 23 patients (20.7%, 23/111). Nineteen (82.6%, 19/23) were subclinical chylothorax cases, which implies a small amount of chyle leakage with no respiratory symptoms. Four (17.4%, 4/23) were clinical, meaning they had either respiratory symptoms, such as dyspnea, desaturation, or a large amount of chylothorax in the images. The incidence was significantly higher in patients who underwent left modified radical neck dissection, and this corresponds to the side in which chylothorax occurred. There were also statistical differences in the drainage color, peak amount, or drain removal time. *Conclusions:* Postoperative chylothorax is a rare complication following neck dissection. However, it can be fatal if the condition progresses. Therefore, patients who undergo total thyroidectomy with modified radical neck dissection, especially on the left side, should be monitored for respiratory symptoms, and serial chest x-ray images obtained.

## 1. Introduction

Differentiated thyroid cancer such as papillary thyroid carcinoma (PTC) is one of the indications for thyroidectomy. The surgical extent is dependent upon the size and invasiveness of the tumors and/or lymph node enlargement. As the extent of surgery increases, the postoperative complications increase as well [1,2]. Chylothorax is a rare but potentially serious complication following neck dissection, with an incidence of less than 1% to 6.2% [1,3,4,5]. However, there are few studies on this complication specifically in total thyroidectomy with neck dissection in PTC patients, and only a few cases have been reported. In advanced thyroid cancer, the risk of chyle leakage is dependent upon the extent of surgical dissection in the neck [3,4,6,7]. Chylothorax is clinically important because it is potentially fatal [8]. However, due to the rarity of cases, few studies have been conducted on the condition and consequently, surgeons are not familiar with its clinical incidence, disease course, and symptoms and signs that allow prompt diagnosis and proper management. The aim of this study is to verify the incidence and clinical characteristics of postoperative chylothorax as a significant complication after total thyroidectomy with modified radical neck dissection.

## 2. Materials and Methods

The records of a total of 111 PTC patients who underwent total thyroidectomy from January 2016 to December 2018 were retrospectively reviewed. The operations were performed by a single surgeon (C.J.Y) at Seoul National University Bundang Hospital, Korea. The medical charts were reviewed for age, sex, surgical approach (open or robotic procedures), the extent of neck dissection, body mass index (BMI), fat-free diet status, postoperative clinical symptoms, the color and amount of drainage, and whether interventions were performed after surgery.

Informed consent was obtained from the patients after thorough discussion and was in accordance with the Declaration of Helsinki and the ethical standards of the Institutional Review Board (B-1711/430-104) on 19 October 2017.

Surgery was performed with open or robotic procedures. Besides flap dissection and a docking step in robotic surgery [9,10,11,12], the main surgical procedures were identical. Depending on the anatomical location and extent of lymph node metastasis, the central and radical neck node dissections were classified as right, left, both, or none if not implemented at all. If only lymph node pickings were performed, the dissection was classified as “other.” The range of modified radical neck dissection (MRND) varied from level II to V, which is from upper jugular region to posterior triangle, depending on the location of enlarged lymph nodes.

In our hospital, after MRND, especially on the left side, it is a routine protocol to insert a Hemovac drain on the side where the MRND was performed. If MRND is done bilaterally, drains are placed on each side. During the operation, the Valsalva maneuver is performed to check for any chyle leaks and if found, the tissues are retracted around the leakage site and closed with sutures. Alternatively, fibrin glue may be used if suturing is difficult. After surgery, a fat-free diet is served and fasting may even be considered depending on the amount of drainage fluid. On the first day after surgery, a chest X-ray is obtained to check for chylothorax, and decisions regarding follow-up imaging studies are made according to the amount of chyle. For clinical chylothorax, the patient’s clinical symptoms are monitored and serial chest X-ray images are obtained. If the patients are determined to be clinically severe cases, we decide to perform interventions such as percutaneous drainage.

Patients who had MRND were given a fat-free diet after surgery and trained to limit their dietary fats for two weeks from the operation day. All patients underwent postoperative chest radiography routinely. The patients were discharged when the drainage amount decreased to under 100 mL/day.

Diagnosis of postoperative chylothorax was determined by: (1) newly occurring pleural effusion compared to the preoperative x-ray (Figure 1, Figure 2, Figure 3 and Figure 4), (2) changes in the color of the Jackson-Pratt (JP) drainage, and (3) the occurrence of new respiratory symptoms after surgery. Among them, the patients with small amounts of chylothorax and no clinical manifestations were defined as the subclinical chylothorax group (i.e., the asymptomatic group), and the patients who had a large amount of drainage with definite clinical symptoms were classified as the clinical chylothorax group (i.e., the symptomatic group).

The descriptive data are presented as mean ± standard deviation or median (range). Statistical analysis was performed with the Student’s *t*-test, Mann—Whitney test, χ2 test, and Fisher’s exact test. A *p*-value of < 0.05 was considered statistically significant. All statistical analyses were performed using R Statistical Software (version 3.4.4; R Foundation for Statistical Computing, Vienna, Austria).

## 3. Results

A total of 111 patients were included in the analysis. Among them, 88 patients (79.3%) had no chylothorax, while the remaining 23 patients (20.7%) had postoperative chylothorax. Table 1 shows the demographics and clinical data of all included patients. The mean age was 45.7 ± 15.1 years in the none group, 45.1 ± 13.1 years in the subclinical, and 40.5 ±14.5 years in the chylothorax group (*p* = 0.568). There was no difference in sex or BMI. Most of the patients had open method surgery with 74 (84.1%) patients in the none group, 13 (68.4%) in the subclinical group, and 4 (100%) in the clinical group. The number of patients who underwent robotic surgery was 14 (15.9%) in the none group, 6 in the subclinical group (31.6%), and none in the clinical group. There was no statistical significance between which approach was performed (*p* = 0.239). Most of the patients underwent central neck dissection. Specifically, bilateral central neck node dissection was dominant, with 82 patients in the none group (93.2%), 19 in the subclinical group (100.0%), and 3 in the clinical group (75.0%). However, there was no statistical difference between the three groups in how much or on which side the central neck dissection was done. There was no difference whether a fat-free diet was undertaken successfully or not after surgery.

Twenty-three (20.7%) patients who had postoperative chylothorax after surgery mostly had chest X-ray abnormalities suggesting postoperative chylothorax, and the severity varied from asymptomatic to severe dyspnea with desaturation. Among them, 19 (82.6%) were classified into the subclinical chylothorax group without respiratory symptoms, and no additional treatment was required. However, one patient in the subclinical group had a large amount of chyle drain which was over 1000 mL/day. Percutaneous drainage intervention was done to prevent further aggravation of patient’s condition, even though the patient did not complain of any symptoms. The other four patients (17.4%) were defined as the clinical chylothorax group, which refers either to those who had pulmonary symptoms and/or a significant amount of chylous pleural effusion requiring additional intervention in two patients (50.0%).

Fifteen patients (78.9%) underwent left MRND in the subclinical group and four (100.0%) in the clinical group, which was significantly more than the other side of lymph node dissection (*p* = 0.019). Regarding clinical manifestations after surgery, 87 (98.9%) patients did not have any symptoms and one (1.1%) patient had only neck swelling in the none group. In the subclinical group, one patient (5.3%) had fever and 12 patients (63.2%) had neck swelling. In the clinical group, no patient experienced neck swelling and 3 patients (75.0%) complained of dyspnea (*p* = 0.005).

There were statistically significant occurrences in left side chylothorax, 13 (68.4%) in the subclinical group and 3 (75.0%) in the clinical group (*p* < 0.001). Only 3 patients (3.4%) had chylous drainage and 85 patients (96.6%) had serous to serosanguineous drain color in the none group. Ten (52.6%) patients in the subclinical group and two (50.0%) patients in the clinical group had chylous drainage postoperatively (*p* < 0.001). There were statistical differences in the average amount of drainage (*p* = 0.008) and the peak amount of drainage (*p* < 0.001).

There was also a statistical difference in the removal time of the drain in the three groups. It took 4.4 ± 2.0 postoperative days to remove the drain in the none group, 6.4 ± 3.5 days in the subclinical group and 6.8 ± 3.0 days in the clinical group (*p* = 0.004).

## 4. Discussion

Until now, few studies of chylothorax after thyroid cancer surgery have been reported [8,13,14]. Before this study, we experienced a severe case of bilateral chylothorax after patient discharge. Due to that event, we routinely obtain a chest X-ray on the first postoperative day for patients who undergo MRND. We found that 20.7% of them had chylothorax. This universal use of imaging as opposed to a case-finding approach may explain the higher detection rate of chylothorax compared to previous literature reports. Cases with only a small amount of chyle with no clinical symptoms were referred to as subclinical chylothorax, and most of them were discharged without any intervention. However, in the case of a large amount of chyle, up to 50% of the patients required interventions. The pathophysiology of chylothorax after MRND can be explained by two mechanisms [4,8,13,14,15,16]. The first mechanism is direct leakage from the base of the neck into the mediastinum as a result of traumatic injury of the thoracic duct (“overflow hypothesis”) (Figure 5) [14,16,17]. This damage of the thoracic duct induces mediastinal tissue maceration and inflammatory reactions, which result in chyle flowing directly into the pleural cavities [8,13]. The other mechanism is that the backward hydrostatic pressure increases after duct ligation, which causes an increase in intraluminal pressure on the duct to create non-traumatic extravasation of chyle (“clogged drain hypothesis”) (Figure 6) [14,17]. This atraumatic leakage of chyle extravasates into one or both pleural spaces, and finally produces chylothorax [18].

It is essential to distinguish postoperative atelectasis or pleural effusion that can occur after general anesthesia from chylothorax through chest X-rays. First of all, thyroid surgery is a relatively simple operation, and the operation time is usually within two to three hours, which brings less chance of respiratory complications [19]. Second, atelectasis can be differentiated from chylothorax by serial X-ray imaging [20]. According to the literature and our experience, cases of pleural effusion after conventional thyroidectomy are exceedingly rare, even in case reports, which suggests that pleural effusions can be ruled out as a common postoperative complication in this setting. In addition, chylothorax can be diagnosed when the color of the aspirated fluid during the intervention is chylous, which is similar to the drainage color coming out of the JP drain. Chylothorax can also be confirmed by measuring fluid triglyceride level [3,4,13,14].

Most of the patients with subclinical chylothorax undergo follow-up chest X-rays at out-patient clinics two weeks after surgery to verify if the chylothorax is resolved. In this study, chylothorax was shown to have resolved in all discharged patients. This finding indicates that subclinical chylothorax does not require any intervention for treatment. However, 50% of the clinical chylothorax patients with a large amount of chyle needed interventions, and one of them needed care in the intensive care unit (ICU) because intubation might have been necessary. That patient underwent nil per oral status and was supported with fluid replacement according to the drainage amount to reduce chyle production [3]. The patient was transferred to the general ward two days after ICU admission. A fat-free diet was carefully administered and, after verifying that the amount of drainage did not increase further, the drain was removed and she was discharged.

The absence of swelling in the neck after left MRND may be a sign that the surgery was successful and implies no leakage of chyle. However, as can be seen from our cases, chyle leakage without neck swelling is still possible when the thoracic duct is ligated and the chyle leakage is into the pleura, causing chylothorax. Therefore, it is important to perform serial chest X-rays to check for postoperative chylothorax even if the patient is asymptomatic.

Despite the originality of this study, there are some limitations that need to be addressed. Firstly, it was a retrospective study, and the number of cases was small because of the rarity of the complication. In addition, patients who did not undergo postoperative chest X-rays were excluded from the study group, further reducing the data power. Secondly, there is insufficient evidence to conclusively confirm that it is postoperative chylothorax just by showing the pleural effusion on X-rays. The chylous appearance of the drainage could be sufficient, but, if additional diagnostic examinations such as drain triglyceride level or thoracentesis are performed, it would further validate the presence of chylothorax.

However, as there is no definite consensus for diagnosis and management, this study may provide evidence for the higher incidence of postoperative chylothorax in MRND patients and differences in clinical manifestations by groups of complications. Moreover, this study also promotes more attention and alertness to the possible complication, which is rare but can be lethal if neglected.

## 5. Conclusions

Postoperative chylothorax is known to be a rare complication following neck dissection. However, the actual incidence rate was higher than expected, especially in the patients who underwent left MRND and it can be fatal if the condition progresses. Therefore, patients who undergo total thyroidectomy with MRND, especially on the left side, should be monitored for respiratory symptoms after surgery, and if possible, serial chest X-ray follow-up is recommended.

## Figures and Tables

**Figure 1 medicina-56-00481-f001:**
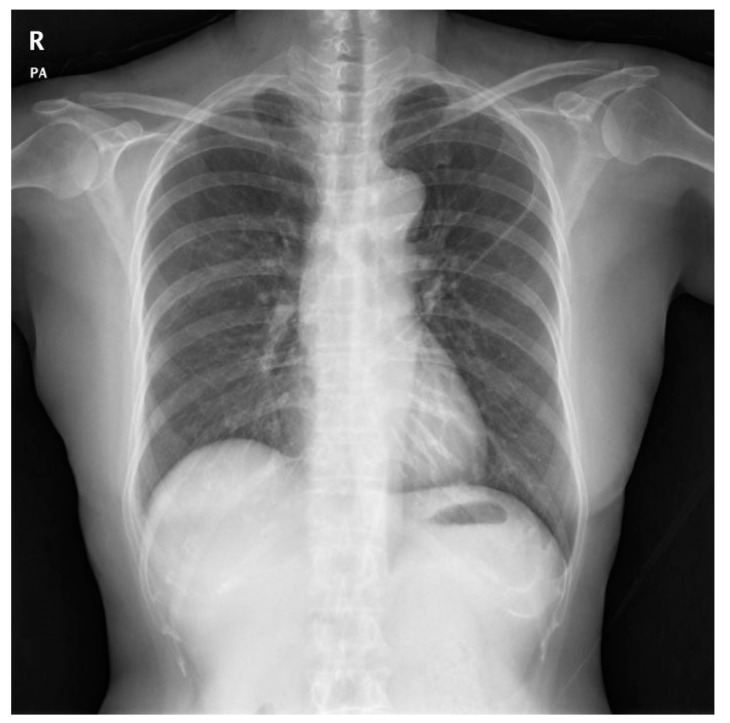
Postoperative chest X-ray with no specific findings (no chylothorax).

**Figure 2 medicina-56-00481-f002:**
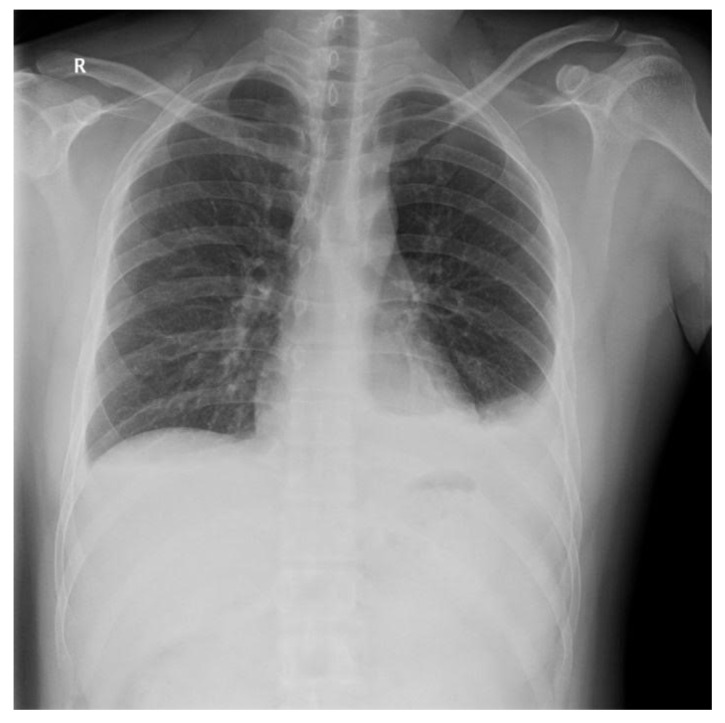
Left side chylothorax after surgery.

**Figure 3 medicina-56-00481-f003:**
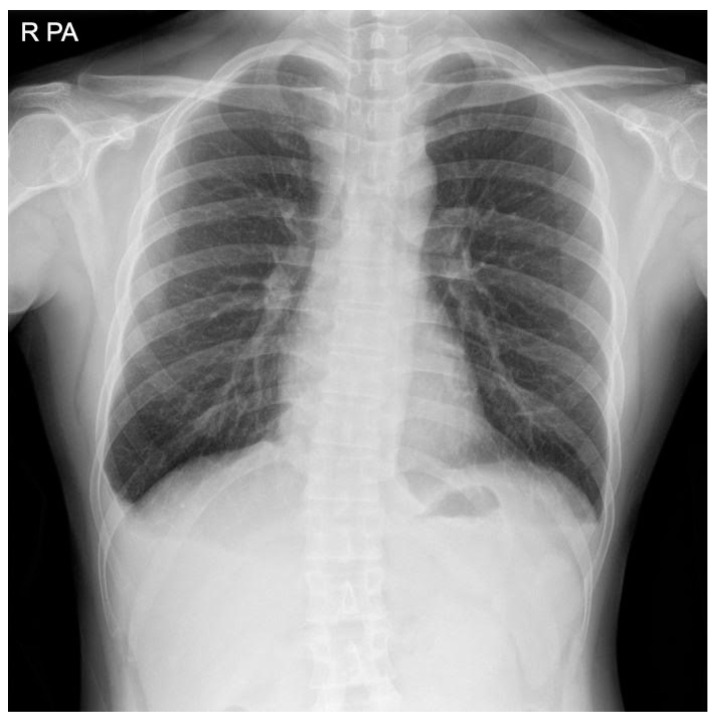
Right side chylothorax after surgery.

**Figure 4 medicina-56-00481-f004:**
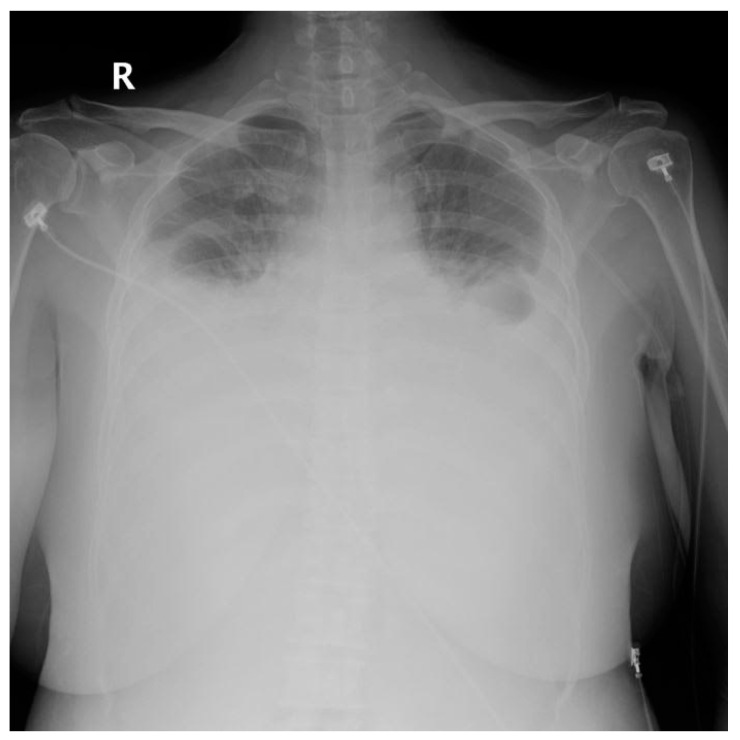
Severe bilateral chylothorax.

**Figure 5 medicina-56-00481-f005:**
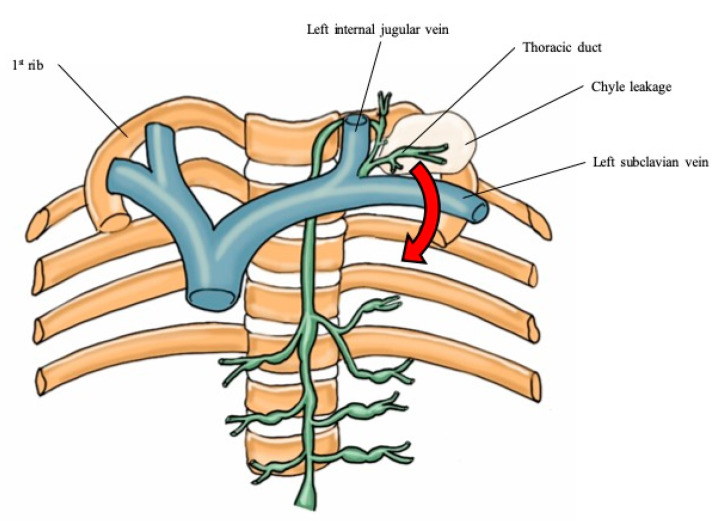
Overflow hypothesis.

**Figure 6 medicina-56-00481-f006:**
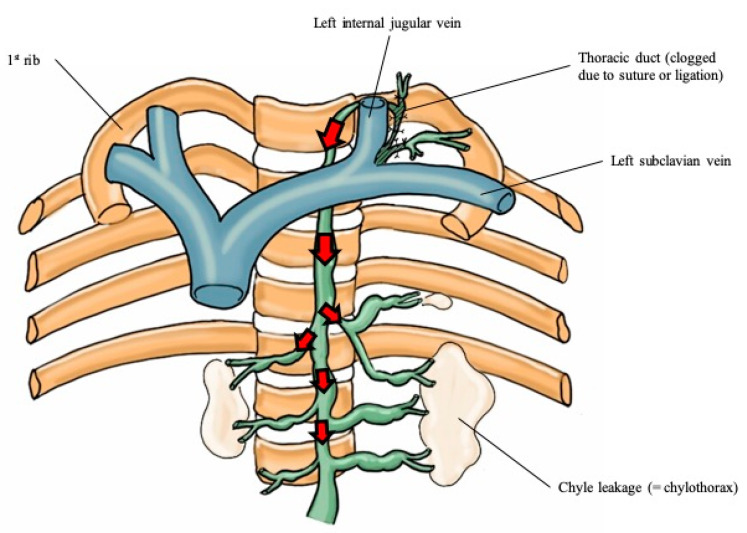
Clogged drain hypothesis.

**Table 1 medicina-56-00481-t001:** Patient demographics and clinical characteristics for all groups. Data are presented as *n* (%) or mean±standard deviation (SD) unless stated otherwise. CND: central neck dissection, MRND: modified radical neck dissection, FFD: fat-free diet.

	None (*n* = 88)	Subclinical (*n* = 19)	Clinical (*n* = 4)	*p*-Value
Sex				0.914
Male	28 (31.8%)	7 (36.8%)	1 (25.0%)	
Female	60 (68.2%)	12 (63.2%)	3 (75.0%)	
Age	45.7 ± 15.1	45.1 ± 13.1	40.5 ±14.5	0.568
Approach				0.239
Open	74 (84.1%)	13 (68.4%)	4 (100.0%)	
Robotic	14 (15.9%)	6 (31.6%)	0 (0.0%)	
CND				0.294
Right	3 (3.4%)	0 (0.0%)	0 (0.0%)	
Left	3 (3.4%)	0 (0.0%)	1 (25.0%)	
Both	82 (93.2%)	19 (100.0%)	3 (75.0%)	
None	0 (0.0%)	0 (0.0%)	0 (0.0%)	
MRND				0.019
Right	20 (22.7%)	1 (5.3%)	0 (0.0%)	
Left	50 (56.8%)	15 (78.9%)	4 (100.0%)	
Both	16 (18.2%)	2 (10.5%)	0 (0.0%)	
Others	2 (2.3%)	1 (5.3%)	0 (0.0%)	
BMI	24.9 ± 3.8	23.1 ± 2.9	23.1 ± 3.0	0.057
FFD				0.273
No	18 (20.5%)	1 (5.3%)	1 (25.0%)	
Yes	70 (79.5%)	18 (94.7%)	3 (75.0%)	
Symptoms				0.005
Dyspnea	0 (0.0%)	0 (0.0%)	3 (75.0%)	
Fever	0 (0.0%)	1 (5.3%)	0 (0.0%)	
Neck swelling	1 (1.1%)	12 (63.2%)	0 (0.0%)	
None	87 (98.9%)	6 (31.5%)	1 (25.0%)	
Site of Pleural effusion				<0.001
Right	1 (1.1%)	2 (10.5%)	0 (0.0%)	
Left	0 (0.0%)	13 (68.4%)	3 (75.0%)	
Both	0 (0.0%)	2 (10.5%)	1 (25.0%)	
None	87 (98.9%)	2 (10.5%)	0 (0.0%)	
Intervention				<0.001
No	88 (100.0%)	18 (94.7%)	2 (50.0%)	
Yes	0 (0.0%)	1 (5.3%)	2 (50.0%)	
Drainage color				<0.001
Chylous	3 (3.4%)	10 (52.6%)	2 (50.0%)	
Serous to Serosanguineous	85 (96.6%)	9 (47.4%)	2 (50.0%)	
Average amount	61.9 ± 26.7	272.9 ± 456.5	172.3 ± 180.7	0.008
Peak amount	115.6 ± 56.8	506.9 ± 772.5	575.2 ± 672.9	<0.001
Drain removal (postoperative days)	4.4 ± 2.0	6.4 ± 3.5	6.8 ± 3.0	0.004

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
