# Peer review of "Postoperative Chylothorax after Modified Radical Neck Dissection for Thyroid Carcinoma: A Missable Rare Complication of Thyroid Surgery"

_medicina, 2020, doi:10.3390/medicina56090481_

Round 1

Reviewer 1 Report

This correspond to a well written and interesting article. 

Congratulations for your effort. 

This represent a really interesting manuscript about an underestimated complications. 

The most important limitations is to consider the diagnosis just based of Chest X-rays. However, is well explained. 

Just:

Some minor grammar mistakes needs to be check. And try to follow the rules about reference across the manuscript. 

- In the line 49, reference 8 is not writing according to the journal rules.

- In the line 175, reference 18.

- In the line 151, you can remove the word patient after the word chylothorax.

Author Response

This correspond to a well written and interesting article.

Congratulations for your effort.

This represent a really interesting manuscript about an underestimated complications.

The most important limitations is to consider the diagnosis just based of Chest X-rays. However, is well explained.

Response: Thank you very much for your detailed and thoughtful comments. It would be great to conduct a follow-up study that also performed additional tests to confirm postoperative chylothorax such as drain serum triglyceride or thoracentesis mentioned in the manuscript.

Just:

Some minor grammar mistakes needs to be check. And try to follow the rules about reference across the manuscript.

- In the line 49, reference 8 is not writing according to the journal rules.

Response: The correction has been made according to the comment.

- In the line 175, reference 18.

Response: The correction has been made according to the comment.

- In the line 151, you can remove the word patient after the word chylothorax.

Response: The correction has been made according to the comment.

Reviewer 2 Report

The article is interesting and well written.

The complication should be related to the histological lymph node involvement, to the extention to the IV level, and the type and histological variant.

The incidence of chylotorax appears to be very high compared to literature reports.

References can be improved and updated.

The inclusion of robotic procedures raises some perplexity in consideration of the fact that they are not present in the current guidelines.

Author Response

The article is interesting and well written.

The complication should be related to the histological lymph node involvement, to the extention to the IV level, and the type and histological variant.

Response: Thank you for your thorough review and your time. Although the extent of modified radical neck dissection (MRND) varied from level II to Vb, many patients underwent level II, III, IV lymph node dissection in this study. It will be interesting to conduct follow-up research on chyle leak and lymph node resection according to the cervical levels, considering even histological results.

The incidence of chylotorax appears to be very high compared to literature reports.

Response: Thank you for clarifying the main conclusion of this study. To make the discussion more solid for the readers, further description has been added according to the comment. (Page 6, line 154-155)

patients who undergo MRND take chest X-rays on the first postoperative day as a routine and we found that 20.7% of them had chylothorax. Therefore, it is thought that the detection rate of chylothorax may be higher compared to literature reports. patients who underwent MRND. Therefore, it is thought that the detection rate of subclinical chylothorax may be higher than in the literature.

References can be improved and updated.

Response: References have been updated. However, there are not many recent articles related to a chyle leak after thyroid cancer surgery.

The inclusion of robotic procedures raises some perplexity in consideration of the fact that they are not present in the current guidelines.

Response: According to American Thyroid Association Statement on Remote-Access Thyroid Surgery, BABA robotic thyroid surgery is described as a competitive method of removing the thyroid gland. It is reported in Paek et al. that the surgical outcomes, including complication and completeness rates, were comparable between BABA robotic surgery and conventional open surgery. Furthermore, in Korea, robotic surgery is highly preferred by patients because of its cosmetic advantages and surgical sophistication.

References:

Berber, E., Bernet, V., Fahey, T. J., 3rd, Kebebew, E., Shaha, A., Stack, B. C., Jr, Stang, M., Steward, D. L., Terris, D. J., & American Thyroid Association Surgical Affairs Committee (2016). American Thyroid Association Statement on Remote-Access Thyroid Surgery. Thyroid : official journal of the American Thyroid Association, 26(3), 331–337. https://doi.org/10.1089/thy.2015.0407

Paek, S.H., Lee, H.A., Kwon, H. et al. Comparison of robot-assisted modified radical neck dissection using a bilateral axillary breast approach with a conventional open procedure after propensity score matching. Surg Endosc 34, 622–627 (2020). https://doi.org/10.1007/s00464-019-06808-9
